# Pathophysiology of Preeclampsia: The Role of Exosomes

**DOI:** 10.3390/ijms22052572

**Published:** 2021-03-04

**Authors:** Keiichi Matsubara, Yuko Matsubara, Yuka Uchikura, Takashi Sugiyama

**Affiliations:** 1Department of Regional Pediatrics and Perinatology, Ehime University Graduate School of Medicine, Toon 791-0295, Japan; 2Department of Obstetrics and Gynecology, Ehime University School of Medicine, Toon 791-0295, Japan; takeyu@m.ehime-u.ac.jp (Y.M.); yuka.itani@gmail.com (Y.U.); sugiyama@m.ehime-u.ac.jp (T.S.)

**Keywords:** exosome, inflammation, placentation, preeclampsia, RNA, trophoblast

## Abstract

The pathogenesis of preeclampsia begins when a fertilized egg infiltrates the decidua, resulting in implantation failure (e.g., due to extravillous trophoblast infiltration disturbance and abnormal spiral artery remodeling). Thereafter, large amounts of serum factors (e.g., soluble fms-like tyrosine kinase 1 and soluble endoglin) are released into the blood from the hypoplastic placenta, and preeclampsia characterized by multiorgan disorder caused by vascular disorders develops. Successful implantation and placentation require immune tolerance to the fertilized egg as a semi-allograft and the stimulation of extravillous trophoblast infiltration. Recently, exosomes with diameters of 50–100 nm have been recognized to be involved in cell–cell communication. Exosomes affect cell functions in autocrine and paracrine manners via their encapsulating microRNA/DNA and membrane-bound proteins. The microRNA profiles of blood exosomes have been demonstrated to be useful for the evaluation of preeclampsia pathophysiology and prediction of the disease. In addition, exosomes derived from mesenchymal stem cells have been found to have cancer-suppressing effects. These exosomes may repair the pathophysiology of preeclampsia through the suppression of extravillous trophoblast apoptosis and promotion of these cells’ invasive ability. Exosomes secreted by various cells have received much recent attention and may be involved in the maintenance of pregnancy and pathogenesis of preeclampsia.

## 1. Introduction

Preeclampsia (PE) is a hypertensive disorder of pregnancy associated with renal and/or liver dysfunction with poor perinatal outcomes, including maternal and neonatal mortality [1]. The etiology of PE has been explained with the “two-stage theory” (Figure 1) [2].

In normal pregnancy, appropriate placentation is critical. Extravillous trophoblasts (EVTs) from the cytotrophoblast (CBT) column of a fertile ovum invade the maternal uterine decidua and myometrium, resulting in spiral arterial remodeling to supply abundant maternal blood to the placental intervillous space for the maintenance of fetal growth. This remodeling results in the loss of vascular smooth muscle, leading to relaxation of the spiral artery and marked dilation of the vessel lumens to maintain abundant blood flow for the embryo.

In PE, however, placental perfusion is reduced in early pregnancy when the invasion of EVTs is disturbed, resulting in reduced spiral artery remodeling. EVTs cannot sufficiently invade the decidua and myometrium, the spiral arteries remain narrow, and placental neovascularization is disturbed. As a result, placentation is shallow, leading to reduced placental perfusion (poor placentation). In mid–late pregnancy, the reduced perfusion with disturbed neovascularization [3], posited as secondary to failed remodeling of the maternal vessels supplying the intervillous space, is not sufficient to maintain normal pregnancy. Reduced placental perfusion and vascular endothelial injury result in the increased production of humoral factors that injure or activate endothelial cells (ECs), including inflammatory humoral factors, in the uteroplacental circulation [4]. These factors are introduced into the systemic circulation and affect many maternal organs. Trophoblast-derived products may cause PE through EC damage and dysfunction [5], the latter additionally resulting in the stimulation of vascular sensitivity to endogenous pressors and vascular permeability [6]. These products can damage the maternal liver, kidney, eyes, brain, blood vessels, and lungs, placing the woman in mortal danger due to potential multiorgan failure. Various humoral factors and modifiers in the peripheral blood have been reported to underlie the pathology of PE, and the clinical manifestation of this condition varies. The pathophysiology of PE is also modified by maternal genetic and environmental factors. To improve the perinatal prognosis for PE, study of its pathogenesis, prediction [4], prevention [7], and treatment is important.

Rolnik et al. [7] reported that the measurement of blood flow in the uterine artery and the concentrations of pregnancy-associated plasma protein A and placental growth factor (PlGF) is useful for PE prediction, and that aspirin (150 mg/day) significantly reduced the incidence of early-onset PE relative to placebo (odds ratio = 0.38) in a high-risk group. However, the pathogenesis of PE needs to be evaluated in more detail to enable more accurate prediction and prevention of this disorder. The identification of women at high risk of PE before its onset is especially important. Recently, microRNAs (miRNAs) and proteins wrapped in an exosome, a subtype of extracellular small vesicle (20–130 nm), were found to be secreted from the placenta into the systemic circulation, resulting in multisystemic organ damage, in patients with PE [8]. As trophoblast-derived exosomes in preeclamptic placentas are thought to have high concentrations of PE-specific contents, the evaluation of exosomes in the peripheral blood of patients with PE is considered to be important. To improve the prognosis, prevention, and treatment of PE, humoral factors, including exosomes, produced from early pregnancy need to be investigated.

## 2. Humoral Factors Related to the Pathogenesis of PE

PE is characterized by chronic inflammation beginning early in pregnancy, with leukocyte activation and high serum levels of cytokines [9,10]. Tumor necrosis factor-α (TNF-α), a monocyte-derived cytotoxic protein, can induce vascular activation and dysfunction via the activation of leukocytes and the induction of vascular endothelial adhesion molecules. In the setting of chronic inflammation, increased TNF-α levels in early pregnancy may stimulate the expression of intercellular adhesion molecule-1 (ICAM-1) on vascular ECs and trophoblasts, thereby activating them. Essentially, ECs mediate vascular homeostasis and regulate the coagulation cascade [11] through the maintenance of vascular tone and permeability [12,13,14]. Chronic inflammation also activates lymphocyte function-associated antigen-1 on leukocytes, resulting in disturbed remodeling of spiral arteries with EC and trophoblast activation. Such activation of ECs leads to vascular dysfunction, resulting in PE (Figure 2).

In normal pregnancy, the fertile ovum is protected by immune tolerance, resulting in no dysfunction of ECs, trophoblasts, and leukocytes, and no inflammation. For this reason, EVTs can invade the uterine decidua and myometrium, resulting in spiral arterial remodeling. To maintain immunotolerance at the fetal–maternal interface, interaction of human leukocyte antigen (HLA)-G with decidual natural killer (NK) cells [15,16,17], between dendritic cells (DCs) and regulatory T (Treg) cells [18], and of cytokines secreted by uterine NK cells [19] in the decidua through reduction of the maternal immune response is needed. However, immune tolerance to fetal antigens, such as trophoblasts, is impaired in PE. Impaired remodeling of the spiral artery could reduce subsequent placentation, resulting in impaired fetal growth due to poor uteroplacental circulation. In the beginning of PE pathogenesis, trophoblasts stimulate the maternal immune response, leading to chronic inflammation. In turn, serum levels of cytokines, especially proinflammatory cytokines such as TNF-α, are significantly increased in early pregnancy by the active immune response; such increases may have predictive value for PE development [20].

Furthermore, in PE, type-1 T helper (Th1) cells are known to be dominant [21] and can secrete proinflammatory cytokines, such as TNF-α, interferon-γ (IFN-γ), and interleukin (IL)-6. Th17 cells also can secrete the proinflammatory cytokine IL-17. The serum TNF-α concentration is reportedly increased even before the onset of PE [20,22].

Chronic inflammation is also related to oxidative stress, an important humoral factor generated by poor placental perfusion in PE [23]. Oxidative stress stimulates the adhesion of leukocytes to the vascular endothelium and the release of cytokines and antiangiogenic factors. Soluble adhesion molecules, such as soluble E-selectin and soluble ICAM-1, are increased in blood collected early in pregnancy from women who subsequently develop PE [20], suggesting that inflammation plays an important role in the first step of PE pathogenesis. The reactive oxygen species level, which is increased during inflammation, was also found to be increased in early pregnancy in women who subsequently developed PE [24].

Abundant angiogenesis at the implantation site in early pregnancy is critical for placental development, with reduced uterine vascular resistance allowing the supply of plenty of maternal blood to the placenta. The disturbance of angiogenesis leads to vascular dysfunction, resulting in poor placentation, and is a main cause of PE [3,4]. Increased levels of antivascular growth factors (e.g., soluble fms-like tyrosine kinase 1 (sFlt-1) and soluble endoglin (sEng)) in the maternal circulation are believed to be important in the pathogenesis of PE, as they reduce angiogenesis in the placenta, and are known to be present even before the onset of PE [25]. Such factors are also involved in the pathogenesis of PE, as they decrease the PlGF level, resulting in poor angiogenesis during placentation.

As many humoral factors are known to be involved in the pathogenesis of PE, the measurement of these factors in the maternal peripheral blood might enable the prediction of PE occurrence. At present, quantification of the serum levels of the antiangiogenic factor sFlt-1 and the angiogenic factor PlGF is the most widespread means of predicting PE development. However, the many methods used currently to predict the occurrence of PE are not adequately sensitive. Recently, exosomes have been identified as important predictive factors for PE; they have been reported to transport various humoral factors (including RNA) to distant organs and are thought to play an important role in PE-related systemic organ damage [21].

## 3. Exosomes

Extracellularly secreted vesicles are known to be involved in cell–cell communication [26], immunity regulation [26,27], and cancer growth [28]. Extracellular vesicles (EVs) have lipid bilayer membranes that surround specific cargos of biomolecules [29]. The membranous vesicles are categorized as exosomes, shedding microvesicles (SMVs), and apoptotic bodies (ABs) based on the modes of biogenesis and release. SMVs are large vesicles ranging from 100–1000 nm in diameter [30] that are released from the plasma membrane through processes such as budding. They are thought to contain high concentrations of matrix metalloproteinases (MMPs) [31], P-selectin [32], and Mac-1 [33], according to cell characteristics. ABs (50–5000 nm in diameter) are heterogeneous vesicles that are released from dying cells, causing apoptosis [34,35]. This process selectively removes aged, damaged, infected, and aberrant cells to maintain healthy tissues; it dismantles cells, and the cellular debris is packed into ABs. Thus, ABs may be involved in cell dismantling and recycling of biomolecular building blocks.

Exosomes (50–100 nm in diameter) are secreted by most cells and are involved in antigen presentation and the transportation of various substances (e.g., messenger RNA (mRNA), miRNA [36], DNA [37], and proteins [38]) to distant organs while avoiding immune response stimulation. They are thought to modify the functions of various organs [39,40]. The exosome membrane is a lipid bilayer that contains cholesterol and sphingomyelin [41] and expresses many types of protein [42].

Although EVTs do not metastasize in the way that malignant tumors do, their autonomous cell proliferation, invasion, and ability to form their own nutrient vessels are similar to those of cancer cells. Cancer cells are known to secrete more exosomes than normal cells [28]. Exosomes can be endocytosed in or interact with recipient cells [43], leading to their involvement in cancer growth and metastasis [44]. Cancer cell–derived exosomes are known to inhibit the function of immune cells that attack cancer cells by causing other cells to act on various substances [45], promoting cancer cell proliferation and vascular EC migration to attract blood into tumors. Yang et al. [45] reported that programmed death ligand-1 (PD-L1)–containing exosomes suppressed immune activity against tumor cells, which suggests that exosomes secreted by cancer cells create an environment conducive to cancer cell growth. The conjugation of PD-L1 and PD-L suppresses T-cell activity by decreasing the production of proinflammatory cytokines [46]. This immune checkpoint signal allows cancer cells to resist the host immune response. Such immune tolerance is also necessary for pregnancy establishment, and the same phenomenon may occur in trophoblasts from the beginning of pregnancy through placentation.

The analysis of exosomes in blood, where they are abundant, is thus a less invasive and sensitive means of diagnosing cancer. This is also true for the analysis of trophoblasts in pregnancy, as these cells proliferate and migrate to create a suitable environment for the fetus. Embryo- and decidua-derived exosomes are thought to contribute to trophoblast proliferation and invasion for embryo implantation. The physiological functions of exosomes are also thought to be important for the maintenance of pregnancy. Placenta-derived exosomes have been detected in the maternal peripheral circulation, and their levels vary in patients with PE [47]. Exosome-based methods might be developed for biomarker analysis and as therapeutic tools for clinical practice in the future.

### 3.1. Roles of RNA and Exosomes during Pregnancy

Fetal cell–free RNA, derived mainly from the placenta, has been detected in maternal plasma [48] and can be used for the noninvasive prenatal examination of physiological and pathophysiological changes in pregnancy, including various forms of placental dysfunction. As the mRNA profile of maternal plasma is related to the placental gene expression profile, it is thought to reflect fetal status [49], and differences in this profile from that in normal pregnancy provide clinical information for the diagnosis of complications such as PE.

miRNAs are small noncoding single-stranded RNAs that regulate gene expression. They consist of about 22 nucleotides and play important roles in intracellular functions. More than 2000 miRNAs, which regulate more than one-third of human genes, have been discovered in the human genome [22]. Pegtel et al. [50] reported that miRNAs were secreted actively through exosomes to protect them from degradation by RNases, suggesting that they function outside of the cells in which they were produced [36].

Czernek et al. [51] reported that exosomes secreted by CBTs, which express placenta-specific miRNAs, including syncytin-2, were involved in embryo implantation via the promotion of Treg differentiation and suppression of the nuclear factor-κB signaling pathway, and thereby the immune reaction and inflammatory response. Devor et al. [52] reported that the exosome miRNA pattern in the first trimester of pregnancy differs between women with PE and those with normal pregnancies, which could be used for early PE diagnosis. Long intergenic/intervening RNAs (lincRNAs) are autonomously transcribed RNAs with more than 200 nucleotides that do not overlap with protein-coding genes [23]. Decidualization is important in the first step of placentation and essential for EVT invasion. Among the many lincRNAs, LINC473 was reported to be involved markedly in decidualization via the regulation of crucial decidual factors and WNT4 [53].

EVs are released from syncytiotrophoblast (STB) membranes into the maternal circulation [54,55], and exosomes released by STBs contain DNA [56]. Fetal cell–free DNA in STB-derived exosomes may play an important role in the physiology of normal pregnancy and can be used for the noninvasive prenatal diagnosis of chromosomal anomalies [57].

### 3.2. Roles of Exosomes in the Pathogenesis of PE

The disturbance of placentation is thought to be a key mechanism underlying PE pathogenesis, and the study of the function of trophoblasts as placenta components is considered to be important to gain an understanding of this pathogenesis [21]. Trophoblasts are classified as STBs, CBTs, and EVTs. STB microvilli derived from the placenta could be pathophysiological markers of PE and are thought to inhibit the proliferation and disrupt the growth of ECs [6]. Exosomes are reportedly released from STBs into the maternal circulation [58]; increased exosome levels were found in preeclamptic women, and they resulted in the endothelial dysfunction underlying the maternal complications that lead to vascular constriction in PE [54]. Neprilysin (NEP), which is highly expressed in the kidney, may be involved in vasoconstriction, sodium retention, and the pathogenesis of PE through the activation of signaling peptides such as endothelin and atrial natriuretic peptide [59,60]. NEP is also widely expressed in placental trophoblasts, which may directly promote impaired uteroplacental circulation. It is reported that active NEP was released into the maternal circulation and bound to STB-derived EVs, and that its expression was increased in PE [61,62]. STB-derived EVs could be the link between placental dysfunction and subsequent PE-related clinical maternal syndromes [63].

Although exosome levels in maternal blood were markedly higher in women with early-onset (at <34 gestational weeks) PE than in those with late-onset (at ≥ 34 gestational weeks) PE [58], the pathogenesis of early-onset PE is considered to conform more closely to the two-stage theory (i.e., to be based on poor placentation) than that of late-onset PE. Thus, exosomes associated strongly with trophoblasts may be more likely to appear in the circulation of pregnant women with early-onset PE, and exosome concentrations and profiles may reflect the PE phenotype. Pillay et al. [8] reported that enrichment in exclusively upregulated exosome RNA in women with PE reflected RNA dysregulation, resulting in abnormal cell growth, adhesion, migration, and invasion. Several researchers have reported on the relationship between exosomes and PE, which may provide new insights into PE diagnosis and treatment [64].

Ermini et al. [65] reported that PE-derived exosomes were involved in vascular dysfunction due to their abundant sFlt-1 and sEng contents. These proteins attenuate EC proliferation, migration, and differentiation, resulting in endothelial dysfunction. Secreted proteomes participate in intercellular signaling, innate immunity, and the construction of extracellular matrix scaffolds around cells [66]. Similarly, placental exosomes may deliver many types of protein around trophoblasts, creating a supportive environment and interfering in the functions of distant organs.

Cancer cells and the surrounding stromal cells that provide such a suitable environment secrete exosomes and are involved in cancer progression. Luga and Wrana [67] reported that fibroblast-derived exosomes promoted the invasion and metastasis of breast cancer cells. Trophoblasts and uterine decidual cells may have a relationship similar to that of cancer cells and stromal cells. The endometrium undergoes desquamation under the influence of luteinizing hormone. Although immune cells such as maternal lymphocytes are present in the decidua, the immune response is reduced and the invasive and proliferative capacities of EVTs are promoted during endometrial decidualization, resulting in placenta formation [68].

Macrophages derived from patients with PE were found to inhibit decidual cell proliferation via apoptosis by proinflammatory cytokines, such as TNF-α and IL-1β [69,70]. Interactions between fetal components and maternal cells, including the actions of exosomes secreted from different cell types on each other, may be related strongly to the maintenance of pregnancy and placentation. Exosomes derived from innate cells contribute to antigen presentation to T cells [71] and the development of tolerance [72]. Luo et al. [73] reported that placenta-specific miRNAs released from placental trophoblasts with exosomes into the maternal circulation may be involved in the regulation of TNF signaling in placental trophoblasts. Exosome-derived miRNAs (e.g., miRNA 517A) released from trophoblasts into the maternal circulation have also been found to be deeply involved in the regulation of the Th1/Th2 balance leading to immune response activation; an imbalance in this context may be responsible for the pathogenesis of PE.

### 3.3. Exosomes Can Be Used as Markers of Pathology

Exosomes contain molecules with characteristics of the cells that secrete them (e.g., cancer cells). The exosomes secreted by trophoblasts at the implantation site are thought to have trophoblast-specific characteristics and to be involved in cell proliferation and invasion. In PE, these exosomes may contain molecules with characteristics of damaged trophoblasts; thus, the analysis of miRNAs, DNA, and proteins contained in exosomes might help to predict the onset of PE. Small changes caused by trophoblast damage cannot be captured due to the interference of many molecules in the analysis of humoral factors derived from peripheral blood. As exosomes have characteristics of disease-causing cells, analysis of the contents of exosomes extracted from maternal blood is considered to be more accurate for the early diagnosis of PE [74]. Currently, PE prediction via the analysis of blood proteins, including sFlt-1, sEng, and PlGF, is only possible just before disease onset; the analysis of blood exosomes may enable much earlier diagnosis.

For the accurate prediction of PE onset, a reliable serum biomarker is essential. Cell-secreted exosomes are thought to contain miRNAs similar to those in the original cells, and cell-derived exosomes from patients with PE may be useful for the prediction of PE development. Although proinflammatory cytokines have been studied as biomarkers of pathophysiology, the miRNAs contained in exosomes are expected to enable more accurate determination of disease prognoses [75]. For example, increased miRNA-215-5p and miRNA-10b-5p levels in serum were associated with poor prognoses for patients with hepatocellular cancer [76]. Intercellular communications encoded by RNA may show signs of cancer and other diseases. Thus, a method based on the detection of miRNAs in exosomes also could be useful for disease diagnosis. Yoshioka et al. [77] developed Exoscreen, a method for the diagnosis of colorectal cancer via the detection of proteins on exosome surfaces. Such analysis by liquid biopsy may become widely used in the future.

The use of exosome-based biomarkers may improve the early diagnosis of PE. Salomon et al. [78] reported that miRNA-486-1-5p and miRNA-486-2-5p levels were elevated in placenta-derived exosomes in the plasma of women with PE. Low concentrations of placental protein 13 in exosomes may be relevant for the early diagnosis of PE, as this protein is involved in early placental development and the regulation of maternal immunoreaction through T-cell and macrophage apoptosis [8,79]. Moreover, exosome-derived syncytin-2, an immunosuppressive molecule, can inhibit the activation of T lymphocytes and NK cells through the Fas ligand and PD-L1 [80,81]. In general, placenta-derived exosomes are synthesized by STBs via exocytosis and released into the maternal circulation. STB-derived exosomes are involved in immunoregulation during pregnancy via the activation of maternal lymphocytes (resulting in the identification of paternal placental antigens) and induction of the apoptosis of semi-allograph invading trophoblasts by exosome-mediated secretion of FasL, contributing to the pathogenesis of PE.

### 3.4. Use of Exosomes in Treatment

In the development of cancer, exosomes are harmful and beneficial to the human body and cancer cells. Exosomes that are harmful to the body promote cancer cell proliferation and immune tolerance to cancer cells, resulting in invasion and metastasis via the regulation of the pericellular microenvironment. These functions also facilitate trophoblast invasion at the implantation site. Like cancer cells, trophoblast-specific exosomes are thought to act on surrounding tissues to create a suitable environment for the semi-allogeneic graft of the fetus. The mother should have a graft-versus-host disease (GVHD)-like reaction to the fetus, which could cause PE or spontaneous abortion, but immune tolerance protects the fetus in normal pregnancy. HLA-G is known to be involved in immune tolerance [82], and mesenchymal stem cell (MSC)-derived exosomes may also be involved. In the context of PE, MSC-derived exosomes are known to inhibit the pathogenesis of GVHD by suppressing more than 50% of the proinflammatory cytokines involved in the pathogenesis of PE, such as IL-1β, TNF-α, and IFN-γ, through the reduction of immunity [83]. They may also be related to the reduction of oxidative stress caused by endothelial ischemia/reperfusion injury [84]. Thus, the effectiveness of MSC-derived exosomes in the treatment or prevention of PE should be investigated. If these exosomes do not create a suitable environment for the fetus in the decidua, placentation may be disturbed, leading to the development of PE.

Exosomes that are beneficial to the body are secreted from MSCs and may be effective in the treatment of various diseases due to their cell proliferation, antiapoptosis, anti-inflammatory, and immunosuppressive effects [85]. The use of patients’ own MSCs can also prevent immune rejection of transplanted organs by reducing the activation of the host defense system. Exosomes produced from MSCs are known to suppress cancer growth. MSCs are used in regenerative medicine, and their pleiotropic effects are reportedly mediated in a paracrine manner [86]. MSC-derived exosomes include miRNAs, mRNAs, cytokines, and growth factors, and may improve the phenotypes of several diseases, such as myocardial infarction, by improving endothelial function and modifying lymphocyte function [84,87]. Thus, MSC-derived exosomes could contribute to the repair of placental vascular dysfunction and chronic inflammation in PE. They were reported to decrease proinflammatory cytokine levels and stimulate Treg cell activity [88] (i.e., reverse the pathogenesis of PE), which can lead to the production of transforming growth factor β (TGF-β) and IL-10 to suppress proinflammatory reactions. Wang et al. [89] reported that miRNA-133b derived from exosomes in umbilical cord blood–derived MSCs stimulated trophoblast cell proliferation, migration, and invasion, thus having a favorable effect in women with PE. Further research to determine whether MSC-derived exosomes are candidate agents for the treatment of PE through cell proliferation, antiapoptosis, anti-inflammatory, and immunosuppressive effects is needed.

STB-derived exosomes might be involved in the pathogenesis of PE, and STB damage may have important effects on exosome secretion in the initial stages of PE pathogenesis. Autophagy may prevent STB cell damage by inhibiting apoptosis, infection, and inflammation [90,91]. As MSC-derived exosomes promote trophoblast proliferation and autophagy under hypoxic conditions, MSCs may inhibit the pathogenesis of PE by decreasing the amount of exosomes released during STB apoptosis by autophagy. Animal experiments and clinical studies are needed to provide insight on the use of MSC-derived exosomes in the treatment of PE.

In addition, the preconditioning of MSCs (e.g., with cytokines) enhances exosome efficacy via immunosuppression and angiogenesis [92]. Gorin et al. [93] reported that MSC-derived exosomes stimulated by fibroblast growth factor (FGF)-2 increased the levels of vascular endothelial growth factor (VEGF) and hepatocyte growth factor, and improved vascularization. Redondo-Castro et al. [94] reported that MSC-derived exosomes decreased the levels of IL-6 and TNF-α. As TNF-α is known to be a key proinflammatory cytokine in the pathogenesis of PE, MSC-derived exosomes may improve the pathophysiological condition in the preeclamptic placenta. Furthermore, Sun et al. [95] reported that platelet-derived exosomes regulated EC proliferation and migration via increased expression of miRNA-126 and angiogenic factors such as VEGF, basic FGF, and TGF-β1. RNA, DNA, and protein-carrying vesicles (e.g., exosomes) may be safe and simple modes of cross-barrier (e.g., blood–brain or blood–placenta) delivery in clinical applications for cardiovascular, neurological, and immunological disorders.

Exosome release from almost all cells in the cardiovascular system may be caused by stressors such as hypoxia and inflammation and may lead to the improvement and repair of cardiac function [96]. Hypoxia induced TNF-α expression in cardiomyocyte-derived exosomes involved in acute myocardial infarction [97]. Bellin et al. [98] reported that exosomes not only bind to target cells and contribute to therapy, but also are useful as biomarkers of cardiovascular disease. Exosomal RNA molecules in cerebrospinal fluid are reliable biomarkers for the differentiation of Parkinson’s disease forms [99], and exosomes carrying small interfering RNA (siRNA) may be useful for the treatment of Alzheimer’s disease (AD) [100]. Rabies viral glycoprotein exosomes are expected to be capable of delivering siRNA specifically and safely for genetic therapies targeting chronic neurodegenerative disorders, including AD [101]. Strong expression of HOX transcript antisense RNA in serum exosomes from patients with rheumatoid arthritis (RA) modulates MMP expression and could be a biomarker for RA diagnosis [102]. DC-derived exosomes with the FasL immunosuppressive ligand may have clinical applications in the treatment of autoimmune diseases, including RA [103].

Rajakumar et al. [104] reported that the levels of placental STB–derived microparticles in the maternal circulation were increased significantly, contributing to systemic maternal vascular injury, in PE. Thus, maternal organ failure in PE may be inhibited by the reduction of the level of exosomes derived from the preeclamptic placenta. The use of exosomes as drug-delivery systems may be also considered for the treatment of PE [105,106]. However, evidence that exosomes are suitable for the treatment of PE is currently insufficient. The functions of exosomes derived from trophoblasts needs to be examined further.

Recent data show that exosomes released from the placenta can cause the systemic pathophysiological changes of PE (Table 1). The study of relevant biochemical, cellular, and molecular mechanisms in an animal model established with exosomes isolated from maternal blood and placental tissue from patients with PE might elucidate the critical roles of these exosomes. Currently, patients with PE are being recruited and patient samples are being obtained for the identification, characterization, and quantification of important functional roles of exosomes in a clinical trial (https://clinicaltrials.gov/ct2/show/NCT04154332; Date of last access: 2 March 2021). Data from this study may aid the development of novel therapeutic intervention strategies for PE.

## 4. Conclusions

Exosomes may be involved in the pathogenesis of PE and have great potential for the treatment of this disease. PE pathogenesis could be ameliorated or prevented by inhibiting exosome effects or preventing their binding to target organs. We would first examine the effects of exosomes released from the preeclamptic placenta on various organs by searching for proteins, RNA, and DNA in the exosomes. Exosomes could be used as markers to predict the onset of PE and to follow the course of this disease [76]. Marleau et al. [107] suggested that the anticancer effects of molecular targeted drugs could be enhanced by removing exosomes from the circulating blood via hemodialysis. This approach may also be used to treat PE via the removal of STB EVs. The use of various exosome-based methods may aid the identification of the best solution for PE prevention and treatment.

## Figures and Tables

**Figure 1 ijms-22-02572-f001:**
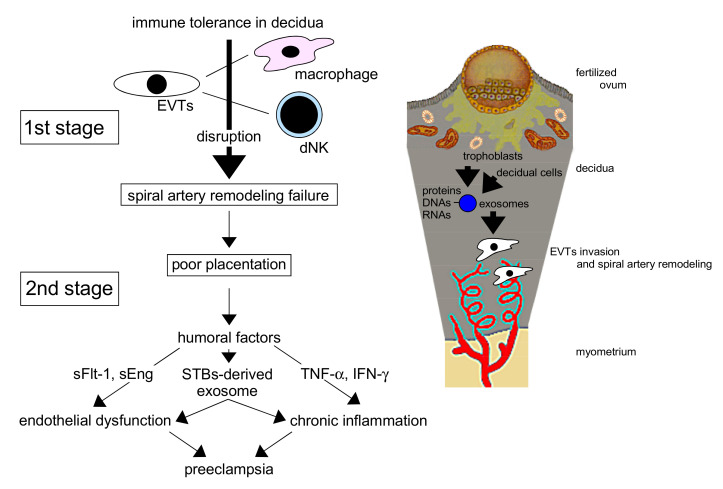
The interaction of decidual cell– and trophoblast-derived exosomes creates a favorable microenvironment for extravillous trophoblasts (EVTs) in the uterine endometrium, which promotes the invasion of EVTs and the remodeling of spiral arteries for adequate placentation in normal pregnancy. Disturbance of the remodeling in the first stage leads to poor placentation, resulting in preeclampsia pathophysiology in the second stage by placenta-derived humoral factors. dNK: decidual natural killer cell, sFlt-1: soluble fms-like tyrosine kinase 1, sEng: soluble endoglin, STBs: syncytiotrophoblasts, TNF-α: tumor necrosis factor-α, IFN-γ: interferon-γ.

**Figure 2 ijms-22-02572-f002:**
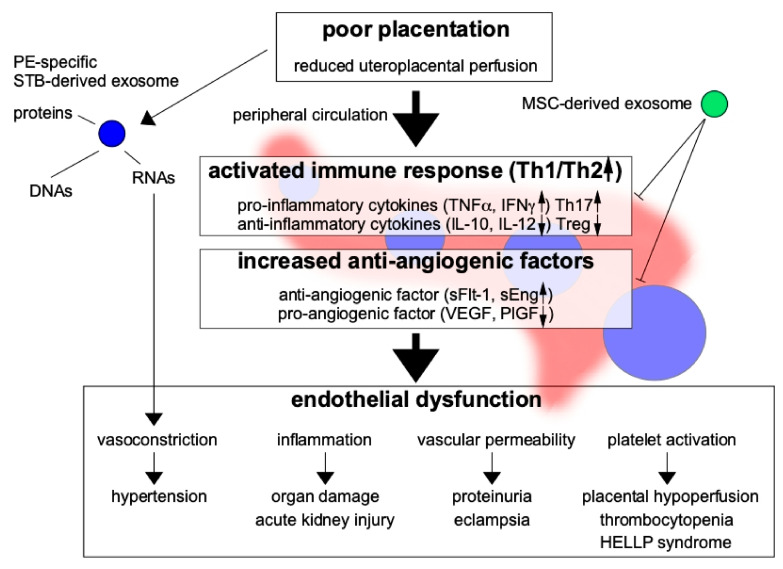
Poor placentation secrets exosomes and promotes immune reactivity and anti-angiogenic factors resulted in increased proinflammatory cytokines and anti-angiogenic factors. On the other hand, anti-inflammatory cytokines and pro-angiogenic factors are decreased. Preeclampsia (PE)-specific exosomes derived from damaged trophoblasts are secreted and may transport RNA, DNA, and proteins to distant maternal organs, causing multiple-organ failure, especially due to endothelial dysfunction. The pathogenesis of PE may be ameliorated by the immunosuppressive and anti-inflammatory effects of mesenchymal stem cell (MSC)-derived exosomes. Th17: type-17 T helper cell, Treg: regulatory T cell, VEGF: vascular endothelial growth factor, PlGF: placental growth factor. Blue circle: pathogenic exosomes, Green circle: exosomes with therapeutic potential.

**Table 1 ijms-22-02572-t001:** Exosomes as potential biomarkers for the diagnosis and treatment of preeclampsia (PE).

Favorable (F) or Harmful (H) for Pregnancy	F	H	Unknown	F	F	F
Type of extracellular vesicle	exosome	extracellular vesicle	exosome	exosome	exosome	exosome
Source	cytotrophoblast	STB	placenta	placenta	UCB-MSC	platelet
Biomarker or treatment	syncytin-2	neprilysin	hsa-miR-486-1(2)-5p	PP13	miR-133b	miR-126, VEGF, basic FGF, TGF-β1
Role or character	immunosuppression of T cells and NK cells	involved in vasoconstriction and sodium retention		involved in early placental development and reduction of maternal immunoreaction	stimulate trophoblast proliferation, migration and invasion	promote EC proliferation and migration
Characteristics as a biomarker		elevated in PE	elevated in PE	decreased in PE		
Advantages or disadvantages on pregnancy	reduce the harmful effects on the embryo	useful for the early diagnosis of PE	useful to identify the risk of developing PE	useful for the early diagnosis of PE	prevent STBs damage by inhibiting apoptosis, infection, and inflammation	therapeutic potential for vascular damage
Reference(s)	[45]	[61]	[78]	[79]	[83,84]	[95]

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
