# Peer review of "Pathophysiology of Preeclampsia: The Role of Exosomes"

_ijms, 2021, doi:10.3390/ijms22052572_

Round 1

Reviewer 1 Report

This articles aims at reviewing the present knowledge about the role of exosomes in preeclampsia (PE). The topic has been recently reviewed by other authors, which  are not cited in the present work (e.g, doi 10.2174/1389200221666200525152441; doi 10.1210/jc.2017-00672; doi 10.3390/ijms21124264)

It should also be mentioned that a clinical trial (https://clinicaltrials.gov/ct2/show/NCT04154332) is recuiting women to isolating  exosomes from maternal blood and placental tissue in patients diagnosed with preeclampsia and studying their biochemical, cellular and molecular mechanism in an animal model.

The article, indeed, is not conveying novel information about the role of exosomes in PE and generates confusion. For example, since STB-derived exosomes are likely involved in PE pathogenesis, MSC-derived exosomes have the opposite effects. Why this is not explained, despite the fact that we are dealing with exosomes in both cases.

As to specific points:

1) Section 1. I would better clarify the "two-stage theory" which is not sufficiently described.

2) Section 2. Lines 84-85: what is the relationship among spiral arterial remodeling and the reduction of the maternal immune response"? This is not clear and appropriate references are not quoted.

3) Section 2. It seems that PE pathogenesis is linked to either inflammation or altered angiogenesis. A schematism providing these information and including also the exosome roles would be useful.

4) Figures 2 and 3 are really dispensable or might be combined in only one.

5) The authors tried to provide a parallelism between the behavior of trophoblast cells with that of cancer cells. While this is interesting, I find that in some places the articles describes just cancer cells (for example, lines 247-264). Moreover, at lines 287-320 only the effects of MSC-derived exosomes are described. In both cases, PE is just on the background. A more synthetic description would aid the readers to better understand that PE is not the focus of such studies.

6) At lines 321-323, the discussion is not on exosomes but on microparticles. Since they are obviously not the same, a preliminary presentation of different cell-derived vesicles would benefit the comprenhension of such lines.

7) All the considerations about the role of exosomes in PE are just presented as hypothetical and scattered throughout the review.

Author Response

This articles aims at reviewing the present knowledge about the role of exosomes in preeclampsia (PE). The topic has been recently reviewed by other authors, which  are not cited in the present work (e.g, doi 10.2174/1389200221666200525152441; doi 10.1210/jc.2017-00672; doi 10.3390/ijms21124264)

Reply: Thank you for your suggestion. I have cited those manuscripts and included additional sentences in red concerning to the manuscripts.

It should also be mentioned that a clinical trial (https://clinicaltrials.gov/ct2/show/NCT04154332) is recuiting women to isolating exosomes from maternal blood and placental tissue in patients diagnosed with preeclampsia and studying their biochemical, cellular and molecular mechanism in an animal model.

Reply: Thank you for your suggestion. We have added some text about the trial. However, this trial has not yet yielded any results, so we cannot write anything definite about it.

The article, indeed, is not conveying novel information about the role of exosomes in PE and generates confusion. For example, since STB-derived exosomes are likely involved in PE pathogenesis, MSC-derived exosomes have the opposite effects. Why this is not explained, despite the fact that we are dealing with exosomes in both cases.

Reply: Thank you for your suggestion. In particular, there were few descriptions of exosomes derived from syncytiotrophoblasts. We added new sentences and new references to P5 and L204.

As to specific points:

  • Section 1. I would better clarify the "two-stage theory" which is not sufficiently described.

Reply: Thank you for your suggestion. We broadened the description of “Two-stage theory” to include more detail and figure 1 was changed.

  • Section 2. Lines 84-85: what is the relationship among spiral arterial remodeling and the reduction of the maternal immune response"? This is not clear and appropriate references are not quoted.

Reply: Thank you for your suggestion. We described in detail and quoted several references.

  • Section 2. It seems that PE pathogenesis is linked to either inflammation or altered angiogenesis. A schematism providing these information and including also the exosome roles would be useful.

Reply: Thank you for your suggestion. We have added a new figure.

  • Figures 2 and 3 are really dispensable or might be combined in only one.

Reply: Thank you for your suggestion. Fig2 and fig3 have been merged into the newly created fig2.

  • The authors tried to provide a parallelism between the behavior of trophoblast cells with that of cancer cells. While this is interesting, we find that in some places the articles describes just cancer cells (for example, lines 247-264). Moreover, at lines 287-320 only the effects of MSC-derived exosomes are described. In both cases, PE is just on the background. A more synthetic description would aid the readers to better understand that PE is not the focus of such studies.

Reply: Thank you for your suggestion. we intentionally divided those descriptions into the pathology section and the treatment section, which seems to have weakened the relevance of each. Therefore, we have added sentences at the end to show exactly the relationship between them.

  • At lines 321-323, the discussion is not on exosomes but on microparticles. Since they are obviously not the same, a preliminary presentation of different cell-derived vesicles would benefit the comprenhension of such lines.

Reply.: Thank you for your suggestion. According to the suggestion, we added several sentences about the role of exosomes on the other diseases.

  • All the considerations about the role of exosomes in PE are just presented as hypothetical and scattered throughout the review.

Reply: Thank you for your suggestion. The role of exosomes in the pathogenesis of PE and their involvement in the treatment of PE has not been established yet, and future research will provide some Replywers on the roles in the future. Therefore, it is inevitable that the sentences are based on hypotheses at this point. However, it is true that the overall story is scattered, and some parts have been revised for clarity.

Reviewer 2 Report

In this manuscript, Matsubara et al. summarize the current knowledge about the cellular and molecular mechanisms involved in the pathogenesis of preeclampsia, and the potential role of circulating exosomes in the development / treatment of this disorder. The manuscript provides a timely and interesting summary on the exosomes in preeclampsia. The paper contains an updated review on published data, as well as a speculative analysis of the topic. However, some concerns must be addressed before publication.

Specific comments

  1. Throughout the manuscript, including the Abstract and Conclusions. While the main focus of the review is preeclampsia, the authors repeatedly discuss findings about cancer and mesenchymal stem cells. In the absence of a clear physiological link with preeclampsia, this is rather confusing. Although the authors partly provide a justification (see for example lines 266-273), the real links are the exosomes related to these cell types. I would suggest focusing on the biological activity of those exosomes and reducing or better explaining/justifying when mentioning “cancer” or “mesenchymal stem cells”. For the sake of clarity and easier reading, the authors should revise the whole manuscript to address this concern.
  2. Exosomes types and activities. Throughout the manuscript, findings about preeclampsia specific exosomes and cancer- or MSCs-derived exosomes are described. However, these exosomes may show opposing functional profiles. While some components of preeclampsia exosomes can display a pathogenic activity, some MSC-derived exosomes show potential therapeutic properties in the preeclampsia treatment. The authors should better illustrate and clarify this subject by adding a Table with the different types of exosomes discussed in the paper, their biological sources, functional activities, therapeutic applications, etc.
  3. The correlative numbering of some sections and subsections needs to be revised. For example, while the section in page 125 is “3. Exosomes”, surprisingly all the following subsections are numbered 2.1 (see lines 157, 183, 230, 265). Also, the following section is numbered “2. Conclusions” (line 331) instead of “4. Conclusions”. Please modify accordingly.
  4. Legend to Figure 2, line 244. “In PE, PE-specific exosomes….” The authors may wish to simplify the legend as follows: “PE-specific exosomes….”
  5. The subsection “Use of Exosomes in Treatment” (line 265). “MSC-derived exosomes could be a new tool for the treatment of PE.” (283,284) “MSC-derived exosomes could be used to repair placental vascular dysfunction and chronic inflammation in PE” (296,297) MSC-derived exosomes may be effective for the treatment or prevention of PE” (lines 283,284)……. All these optimistic statements must be softened by discussing the limitations of the actual exosome-based therapy. Although exosomes derived from multipotent mesenchymal stromal cells are currently being tested in clinical trials, their efficacy needs to be substantiated.
  6. Line 202,203. “On the other hand, Chang et al. [45] reported that PE-derived exosomes were involved in vascular dysfunction due to their abundant sFlt-1 and sEng contents” The authors should cite and comment the original article describing the presence of sEng in exosomes from preeclamptic women: Ermini et al. Sci. Rep. 2017; 7(1): 12172. doi: 10.1038/s41598-017-12491-4.

Author Response

In this manuscript, Matsubara et al. summarize the current knowledge about the cellular and molecular mechanisms involved in the pathogenesis of preeclampsia, and the potential role of circulating exosomes in the development / treatment of this disorder. The manuscript provides a timely and interesting summary on the exosomes in preeclampsia. The paper contains an updated review on published data, as well as a speculative analysis of the topic. However, some concerns must be addressed before publication.

Specific comments

  1. Throughout the manuscript, including the Abstract and Conclusions. While the main focus of the review is preeclampsia, the authors repeatedly discuss findings about cancer and mesenchymal stem cells. In the absence of a clear physiological link with preeclampsia, this is rather confusing. Although the authors partly provide a justification (see for example lines 266-273), the real links are the exosomes related to these cell types. I would suggest focusing on the biological activity of those exosomes and reducing or better explaining/justifying when mentioning “cancer” or “mesenchymal stem cells”. For the sake of clarity and easier reading, the authors should revise the whole manuscript to address this concern.

Reply: Thank you very much for your suggestion. The most important part in the pathogenesis of PE is impaired migration of trophoblastic cells. Since trophoblastic cells are physiological cancer cells with strong proliferative and invasive capacity, we cited many papers on cancer cells in order to understand the physiological behavior of trophoblasts and the pathogenesis of PE. In addition, one of the major causes of trophoblastic migration disorder is dysfunction of trophoblastic cells due to activated maternal immunity. Therefore, we thought that the immunosuppressive function of MSC-derived exosomes would play an important role in the prevention and treatment of PE in the future, and we devoted many pages to MSCs. However, as the reviewer described, it is undeniable that the number of stories specific to PE is decreased. Therefore, we have reviewed and revised the overall text.

  1. Exosomes types and activities. Throughout the manuscript, findings about preeclampsia specific exosomes and cancer- or MSCs-derived exosomes are described. However, these exosomes may show opposing functional profiles. While some components of preeclampsia exosomes can display a pathogenic activity, some MSC-derived exosomes show potential therapeutic properties in the preeclampsia treatment. The authors should better illustrate and clarify this subject by adding a Table with the different types of exosomes discussed in the paper, their biological sources, functional activities, therapeutic applications, etc.

Reply. Thank you very much for your suggestion. We followed that suggestion and made a table to add.

  1. The correlative numbering of some sections and subsections needs to be revised. For example, while the section in page 125 is “3. Exosomes”, surprisingly all the following subsections are numbered 2.1 (see lines 157, 183, 230, 265). Also, the following section is numbered “2. Conclusions” (line 331) instead of “4. Conclusions”. Please modify accordingly.

Reply. Verry sorry. It was correctly changed.

  1. Legend to Figure 2, line 244. “In PE, PE-specific exosomes….” The authors may wish to simplify the legend as follows: “PE-specific exosomes….”

Reply: Thank you for your suggestion. We changed it simply.

  1. The subsection “Use of Exosomes in Treatment” (line 265). “MSC-derived exosomes could be a new tool for the treatment of PE.” (283,284) “MSC-derived exosomes could be used to repair placental vascular dysfunction and chronic inflammation in PE” (296,297) MSC-derived exosomes may be effective for the treatment or prevention of PE” (lines 283,284)……. All these optimistic statements must be softened by discussing the limitations of the actual exosome-based therapy. Although exosomes derived from multipotent mesenchymal stromal cells are currently being tested in clinical trials, their efficacy needs to be substantiated.

Reply: Thank you for your suggestion. I think you're right. I changed the wording.

  1. Line 202,203. “On the other hand, Chang et al. [45] reported that PE-derived exosomes were involved in vascular dysfunction due to their abundant sFlt-1 and sEng contents” The authors should cite and comment the original article describing the presence of sEng in exosomes from preeclamptic women: Ermini et al. Sci. Rep. 2017; 7(1): 12172. doi: 10.1038/s41598-017-12491-4.

Reply: Thank you for your suggestion. We cited it and changed the sentence.

Reviewer 3 Report

Congratulations to the authors for the submitted manuscript in which you presented the pathophysiological processes in preeclampsia in a very precise way. You presented the connection of classic biomarkers with new ones and offered the reader new possibilities with the introduction of new diagnostic methods. 

Author Response

Thank you for reading this review. I would like to continue to strive doing further research and publishing manuscripts that will meet your expectations.
Thank you very much.

Round 2

Reviewer 1 Report

Although the authors have answered to my concerns, there is still some uncertainty about the classification of different extracellular vesicles, thus a small paragraph describing exosomes, microparticles and apoptotic bodies, accordingly with the International Society for Extracellular Vesicles, should be added. 

I could not find any reference to the clinical trial I mentioned in my previous report.

Author Response

Although the authors have answered to my concerns, there is still some uncertainty about the classification of different extracellular vesicles, thus a small paragraph describing exosomes, microparticles and apoptotic bodies, accordingly with the International Society for Extracellular Vesicles, should be added.

Reply: Thank you for your suggestion. I have become a member of the International Society for Extracellular Vesicles. I referred to the conference website and added a small paragraph describing exosomes, microparticles and apoptotic bodies, citing the paper by Kalra et al (line 147 - line 162).

I could not find any reference to the clinical trial I mentioned in my previous report.

Reply: Thank you for your suggestion. We mentioned it on the line 436 – line 443.

Also, the manuscript has been proofread by a native English speaker again.

Reviewer 2 Report

The manuscript has been improved. However, some minor points remain to be addressed before publication:

-Editing of English language and style is required 

-Table 1. It is dificult to read the text. Please improve its visibility/resolution. 

-Throughout the manuscript, several abbreviations for microRNA  (miRNA, miR, MIR) are used. Please unify for consistency

Author Response

Editing of English language and style is required

Reply: Thank you for your suggestion. The manuscript has been proofread by a native English speaker again.

Table 1. It is difficult to read the text. Please improve its visibility/resolution.

Reply: Thank you for your suggestion. The table 1 was changed to improve the visibility and resolution.

Throughout the manuscript, several abbreviations for microRNA  (miRNA, miR, MIR) are used. Please unify for consistency

Reply: Thank you for your suggestion. The abbreviation has been unified to miRNA.
